

# Climate Induced Speleothem Radiocarbon Variability on Socotra Island from the Last Glacial Maximum to the Younger Dryas

Steffen Therre[1], Jens Fohlmeister[23], Dominik Fleitmann[4], Albert Matter[5], Stephen J. Burns[6], Jennifer Arps[1], Andrea Schröder-Ritzrau[1], Ronny Friedrich[7], Norbert Frank[1]

[1]Institute of Environmental Physics, Heidelberg University, Heidelberg, Germany
[2]Institute of Earth and Environmental Science, University of Potsdam, Potsdam, Germany
[3]GFZ German Research Centre for Geosciences, Section Climate Dynamics and Landscape Development, Potsdam, Germany
[4]Department of Environmental Sciences, University of Basel, Basel, Switzerland
[5]Institute of Geological Sciences, University of Bern, Bern, Switzerland
[6]Department of Geosciences, University of Massachusetts, Amherst, USA
[7]Curt-Engelhorn-Center Archaeometry gGmbH, Mannheim, Germany

*Correspondence to*: Steffen Therre (stherre@iup.uni-heidelberg.de)

**Abstract.** In this study, the dead carbon fraction (DCF) variations in stalagmite M1-5 from Socotra Island in the western Arabian Sea were investigated through a new set of high-precision U-series and radiocarbon ($^{14}$C) dates. The data reveal an extreme case of very high and also climate dependent DCF values. For M1-5 an average DCF of 56.2 ± 3.4 % is observed between 27 and 18 kyr BP. Such high DCF values indicate a high influence of aged soil organic matter (SOM) and nearly completely closed system carbonate dissolution conditions. Towards the end of the last glacial period decreasing Mg/Ca ratios suggest an increase in precipitation which caused a marked change in the soil carbon cycling as indicated by sharply decreasing DCF. This is in contrast to the relation of soil infiltration and reservoir age observed in stalagmites from temperate zones. For Socotra Island, which is influenced by the East African–Indian monsoon, we propose that more humid conditions and enhanced net-infiltration after the LGM led to denser vegetation and thus lowered the DCF by increased $^{14}CO_2$ input into the soil zone. The onset of the Younger Dryas (YD) is represented in the record by the end of DCF decrease with a sudden change to much higher and extremely variable reservoir ages. Our study highlights the dramatic variability of soil carbon cycling processes and vegetation feedback on Socotra Island manifested in stalagmite reservoir ages on both long-term trends and sub-centennial timescales, thus providing evidence for climate influence on stalagmite radiocarbon. This is of particular importance for studies focussing on $^{14}$C calibration and atmospheric reconstruction through stalagmites which relies on largely climate independent soil carbon cycling above the cave.

## 1 Introduction

Radiocarbon ($^{14}$C hereinafter) dating has fundamentally changed our knowledge on the timing of events and rates of change in archaeological and climate proxy records ever since it was pioneered in the late 1940s (Arnold and Libby, 1949). To provide more accurate data for $^{14}$C calibration beyond the tree ring record, speleothems gained importance as a source of information





on atmospheric $^{14}$C. Stalagmite records from tropical and temperate climate zones overlapping the (floating) tree-ring data (Beck et al., 2001; Hoffmann et al., 2010; Southon et al., 2012) have been included in the most recent comprehensive $^{14}$C intercalibration data set IntCal13 (Reimer et al., 2013) . Radiocarbon calibration based on speleothems relies on the assumption

of a low and constant dead carbon fraction (DCF) in the stalagmite, even though one of the included calibration records showed more than 50 % reduction of the reservoir age during the YD (Beck et al., 2001; Hoffmann et al., 2010). A unique speleothem calibration record was recently published by Cheng et al., 2018 who presented a continuous atmospheric $^{14}$CO$_2$ reconstruction from stalagmites with a very low (<6 %) and constant DCF. They further showed that variations of speleothem $^{14}$C at Hulu Cave, China, coincide with floating tree-ring $^{14}$C variations, proving that even small decadal-scale carbon cycle changes and

atmospheric $^{14}$C are captured by stalagmites (Adolphi et al., 2017). However, climatic fluctuations, variations of the carbon cycle and carbonate dissolution on DCF complicate the calculation of atmospheric $^{14}$C from stalagmites. By combining $^{14}$C and elemental Mg/Ca analysis a connection between precipitation/net-infiltration and stalagmite reservoir age was revealed, and is most likely related to changes in carbonate dissolution systematics and the contribution of very old host rock carbonate to drip water dissolved inorganic carbon (DIC) (Griffiths et al., 2012; Noronha et al., 2014). In a recent study DCF values

exceeding 50 % in a Holocene stalagmite from Corchia Cave, Italy have been reported (Bajo et al., 2017). Such high DCF values are above the theoretical limit for closed system carbonate dissolution systems and were attributed to enhanced host rock dissolution by sulphuric acid. Two studies also focussed on the direct influence which aging decomposing soil organic matter (SOM) has upon soil CO$_2$ and subsequently reservoir ages, leading to depleted soil gas $^{14}$CO$_2$ concentration and consequently higher DCF in the stalagmite calcite (Fohlmeister et al., 2011; Rudzka et al., 2011).

The Arabian Sea and Indian Ocean have been the focus of numerous paleoclimate studies in recent years (DiNezio et al., 2018; Fleitmann et al., 2003, 2007; Shakun et al., 2007) particularly with respect to the potential impact of climatic changes in the North Atlantic region on the intensity of the Indian summer monsoon and low-latitude climate conditions. For this study we selected a speleothem (sample M1-5) from Socotra Island, which documents a significant increase in monsoonal rainfall during the Bølling-Allerød (Shakun et al., 2007). These findings were recently corroborated by DiNezio et al., 2018 who have

postulated a unique glacial climate dipole behavior of the Indian Ocean climate linked to the monsoon. This dipole pattern, which seems predominantly driven by the exposure of the Sunda and Sahul shelves, causes a marked cooling and reduction in moisture availability in the Arabian Sea, leading to dry conditions in the region around Socotra Island during the last glacial period.

The investigated stalagmite from Socotra Island provides an opportunity to observe the shifting climate regimes under the

influence of the Indian monsoon starting from temperate conditions in the last glacial period. As the climate shifted towards more humid conditions synchronous changes in vegetation and soil carbon cycling may be expected. Hence, we investigate the temporal evolution of the reservoir age, DCF, and hydro-proxies of stalagmite M1-5 from the LGM to the beginning of the Holocene to study the links between hydro-climate proxies and soil carbon cycling. Our study shows that increasing moisture on Socotra Island goes along with decreasing DCF and proves that the input of $^{14}$CO$_2$ into the soil zone through denser

vegetation above the cave is the determining factor for soil carbon cycling.



## 2 Site and Sample

### 2.1 Socotra Island

Socotra Island is the largest island of the eponymous archipelago of four islands on the western margin of the Arabian Sea. It is situated 230 to 360 km east of the Horn of Africa (Somalia) and 380 km south of the Arabian Peninsula (**Fig. 1**). Socotra Island is approximately 130 km long and 30 km wide. Its topography varies from coastal plains in the North and South of the island to elevated plateaus several hundred meters above sea level (m a.s.l.). The innermost part of the island consists of a mountainous area with elevations up to 1500 m a.s.l. The archipelago is mainly made up of Precambrian basement rocks that are in part overlain by Cretaceous or Tertiary plateau limestone (Schlüter, 2006). The maritime climate of Socotra Island is under strong influence of the East African–Indian monsoon system reflected in the bi-annual migration of the ITCZ between its Northern Hemisphere winter position in the southern Indian Ocean and its summer position in the Arabian Sea (Shakun et al., 2007) (**Fig. 1a, b**). This migration is responsible for the bimodal distribution of precipitation. More than 20 % of total annual rainfall on the island occurs in April/May when the northward migrating ITCZ crosses the island. The second annual crossing of the ITCZ during the time October to December accounts for more than 50 % of the annual rainfall (Scholte and De Geest, 2010). Today, total precipitation can be as low as 67 mm per year near the coastal plains, while more than 500 mm rainfall can occur in the mountainous areas in the inner island with elevations of more than 1000 m a.s.l. (Scholte and De Geest, 2010).

### 2.2 Stalagmite M1-5

Stalagmite M1-5 was collected from Moomi Cave in 2002, which lies on a limestone plateau (Moomi Plateau, c. 500m a.s.l.) in the eastern part of Socotra Island (12°32'04'' N, 54°19'00'' E, **Fig. 1c**). Moomi Cave is approximately 1 km long, mainly horizontal and overlain by ~20m of bedrock (Burns, 2003). Stalagmite M1-5 is approximately 2.2 m long and 10 to 15 cm wide, and grew roughly 1 km from the cave entrance. It was sectioned along its growth axis down to the base (**Fig. 2**) and polished to make the growth layers more visible (Shakun et al., 2007). On the macroscopic scale there are significant differences visible in the texture and stratigraphy of the stalagmite. The lower part of the stalagmite from its base to approximately 1.5 m distance from top (dft) shows distinct changes in layer coloring and generally has a larger width of approximately 15 cm. In younger parts, the carbonate is brighter and the width rarely exceeds 10 cm. Another visible change can be seen in the uppermost 10 cm where the layering is much darker, which Shakun et al., 2007 found to occur shortly after an apparent hiatus.



## 3 Methods

### 3.1 U-series Dating

95   Forty samples were analyzed at the Heidelberg Institute of Environmental Physics for this study. M1-5 was previously U-
series dated at the Isotope Geology Laboratory, University of Bern. In total 62 age data points are presently available, in
addition to the previous age determination efforts by Shakun et al., 2007. For the sample treatment in Heidelberg approximately
to 120 mg of speleothem calcite were cut with a diamond wire saw or drilled along growth layers. Chemical sample
preparation, U and Th purification, and mass spectrometric measurements follow the procedures described in detail in (Arps,
2017; Douville et al., 2010; Wefing et al., 2017). Activity ratios were determined using a Thermo Fisher Neptune Plus multi-
collector inductively coupled plasma mass spectrometer (MC-ICP-MS) at the Institute of Environmental Physics, Heidelberg
University, Germany. Multiple measurements of the HU-1 standard resulted in a value of $1.00002 \pm 0.00082$ for ($^{234}$U/$^{238}$U)
and $1.00004 \pm 0.00082$ for ($^{230}$Th/$^{238}$U) for 517 samples over 22 months. The HU-1 standards used to bracket the sample
measurements accordingly yield a reproducibility of 0.82 permil for ($^{234}$U/$^{238}$U) and ($^{230}$Th/$^{238}$U). Full procedural blanks were
measured smaller than 0.04 fg for $^{230}$Th and 0.4 fg for $^{234}$U. The calculations for the radiometric ages were performed using
the half-lives from (Cheng et al., 2000) to maintain comparability with the record from (Shakun et al., 2007). More recent half-
life values do not significantly change radiometric age estimates (see results section). For detrital $^{230}$Th correction a $^{232}$Th/$^{238}$U
weight ratio of $3.8 \pm 1.9$ and secular radioactive equilibrium within the uranium decay chain was assumed.

### 3.2 Radiocarbon Dating

For $^{14}$C measurements, approximately 10 mg of calcite were cut with a diamond wire saw or drilled from the stalagmite along
the growth layers. To avoid contamination by ambient air, the cut samples were leached in 4 % hydrochloric acid before
hydrolysis, whereas the powder from drilled samples was immediately processed after sampling. Hydrolysis was carried out
on an evacuated glass line ($p < 10^{-3}$ mbar) by adding 0.5 ml of 11 % hydrochloric acid. The emerging water vapor during
hydrolysis was removed by dry ice/acetone freezing traps. The acquired $CO_2$ was subsequently reduced to graphite over 3-4
hours at 575°C on a separate setup by adding $H_2$ and the $CO_2$ from the previous step to a reactor containing iron powder as
catalyst. Detailed descriptions of the sample preparation routine can be found in (Fohlmeister et al., 2011) and (Unkel, 2006).
After the reaction, the resulting graphite-iron compound was pressed into targets. A 200kV tandem AMS of type MICADAS
at Curt-Engelhorn-Center Archaeometry in Mannheim, Germany was used for the measurements (Kromer et al., 2013; Synal
et al., 2007). Process blanks from marble were prepared using the same method applied for regular samples to account for
possible contamination during chemical preparation or sample handling before the eventual measurement in the AMS and
yielded apparent blank activities consistently lower than 0.25 pmC (apparent blank $^{14}$C ages 48–56 kyr). Oxalic acid-II
standards were graphitized from previously extracted $CO_2$ gas and used for measurement calibration. Other international and
in-house calcite standards were prepared and measured frequently to determine the full analytical measurement reproducibility.



In our case the IAEA C2 standard was measured over a period of two years with a reproduced value of 40.77 ± 0.36 pmC
compared to the assigned literature value of 41.14 ± 0.03 pmC.

### 3.3 Elemental Analysis

For measurements of Mg/Ca ratios, approximately 1 to 2 mg of calcite was drilled from the stalagmite at each [14]C sample
point. At the Institute of Environmental Physics at Heidelberg University a Thermo Fisher iCAP Q Inductively Coupled Plasma
Quadropole MS (ICPQMS) was used, while data was also collected at the Institute of Earth Sciences at Heidelberg University
using an ICP optical emission spectrometer (ICP-OES) 720. For the ICPQMS, the 1σ- reproducibility of the Mg/Ca ratio of
two measured standards is <1 %. For the OES setup, the internal 1σ-standard deviation is <1 % for $Ca^{2+}$ and $Mg^{2+}$. Here SPS
SW2 is used as a standard, and the long-term 1σ-reproducibility is 2.0 % for $Ca^{2+}$ and 3.3 % for $Mg^{2+}$ (Warken et al., 2018).
Comparability of the results from both setups was ensured by normalizing the Mg/Ca ratios with respect to a series of internal
standard measurements that were carried out in both laboratories.

### 4 Results

#### 4.1 U-series Chronology

The U concentration in M1-5 ranges from 700.5 ± 2.9 to 2913.8 ± 9.3 ppb, [232]Th concentrations are between 15.04 ± 0.88 and
21042 ± 36 ppt.

All U-series ages in this study are reported in "years before present (yr BP)" or "kilo years before present (kyr BP)" where BP
is referring to the year 1950. High [238]U and low [232]Th concentration in most samples allow for very precise U-series ages
without major corrections or age reversals. U-series ages range from 27.11 ± 0.20 kyr BP to 10.890 ± 0.034 kyr BP with 2σ-
age uncertainties from 0.2 to 1.0 %. The resulting chronology is in good agreement with previous results from the same sample
(Shakun et al., 2007), but the higher resolution of the new U-series dates permits to reveal the growth history in much closer
detail. With the exception of two outliers almost all U-series ages are in stratigraphic order and were used for the age-depth
modeling (**Fig. 3**). In addition, we replicated U-series ages for seven different depths which agree within their 2σ-errors.
Age-depth modeling was performed using stepwise implementation of StalAge (Scholz and Hoffmann, 2011) between the
resolved growth discontinuities. The growth rate of stalagmite M1-5 is fairly constant during the last glacial termination and
interstadials from approximately 17.2 kyr until 11.7 kyr BP at an average growth rate of roughly 0.23 mm yr[-1]. A distinct 800
yr hiatus was identified at 110 mm (from top), from 11.7 to 10.9 kyr BP (henceforth referred to as discontinuity "D1"). Above
this hiatus, the stalagmite growth continued at a high rate of approximately 0.30 mm yr[-1] for a short period at the beginning of
the Holocene. Dating results obtained by (Shakun et al., 2007) as well as the rapid texture and color changes of the growth
layers in the uppermost part of the stalagmite above 5 cm from top indicate sporadic growth throughout the Holocene. For the
older part of the stalagmite a lower growth rate is observed: From 27.2 to 17 kyr BP, the average rate is 0.086 mm yr[-1] with
another 1200 year-long hiatus after 23.5 kyr BP (discontinuity "D2"). Overall, the growth is mostly continuous with two



discontinuities of multi-centennial duration with an overall rate of approximately 0.130 mm yr$^{-1}$ between 27.2 and 11 kyr BP implying a high temporal resolution of 7.5 yr mm$^{-1}$.

The initial $\delta^{234}$U at the time of deposition ($\delta^{234}U_{init}$) can be calculated from the measured $\delta^{234}U_{meas}$ by correcting it for $^{234}$U decay with the determined age t:

$$\delta^{234}U_{init} = \delta^{234}U_{meas} \cdot \exp(\lambda_{234} \cdot t) \tag{1}$$

$\delta^{234}U_{init}$ of stalagmite M1-5 is consistently below zero with minimal values of around -130 ‰, which implies a strongly leached and subsequently $^{234}$U-depleted host rock. $\delta^{234}U_{init}$ increases from the $^{234}$U-depleted LGM ~21 kyr BP towards the Bølling-Allerød warm period (-25 ‰ ~15 kyr BP) where the growth rate is significantly higher.

**4.2 Radiocarbon Results**

A total of 78 stalagmite $^{14}$C measurements were performed on stalagmite M1-5, including 6 duplicates. The measured $^{14}$C

activities ($a^{14}C_{meas}$) range between 19.52 ± 0.20 pmC and 2.846 ± 0.035 pmC. Reproducibility of $^{14}$C results is assessed by long-time measurements of the international IAEA C2 standard. The resulting statistical error is used as overall reproducibility. The total uncertainty is then calculated as the square root of the squared sum of the statistical error and overall reproducibility. Each sample was assigned an absolute age ($t_{mod}$) from the StalAge model, revealing a generally decreasing $^{14}$C activity for older parts of M1-5 but also large variations on decadal timescales. For comparison with atmospheric $^{14}$C levels, the data is

converted to radiocarbon years ($t_{14C}$).

$$t_{14C} = -8033 \text{ yr} \cdot \ln(a^{14}C_{meas}) \tag{2}$$

The $^{14}$C ages of stalagmite M1-5 are significantly higher than the contemporaneous atmospheric $^{14}$C calibration curve (Reimer et al., 2013) and vary substantially, which can be seen in comparison to the Hulu Cave speleothem record (Cheng et al., 2018) (**Fig. 4a**). Based on the StalAge model for M1-5, the initial $^{14}$C activity ($a^{14}C_{init}$) at the time when the respective stalagmite

layer was deposited was reconstructed.

$$a^{14}C_{init} = a^{14}C_{meas} \cdot \exp(t_{mod}/8267 \text{ yr}) \tag{3}$$

DCF was calculated by comparing each $a^{14}C_{init}$ to the atmospheric $^{14}$C level ($a^{14}C_{atm}$) at the respective time, obtained from (Reimer et al., 2013). 1σ-errors were calculated using a Monte Carlo approach following the procedure described in (Griffiths et al., 2012).

$$DCF = \left[1 - \frac{a^{14}C_{init}}{a^{14}C_{atm}}\right] \cdot 100 \text{ \%} \tag{4}$$

DCF values for stalagmite M1-5 range from 27.33 ± 0.24 % at approximately 13 kyr BP to values as high as 64.6 ± 1.1 % at 20.7 kyr BP (**Fig. 4b**). The M1-5 DCF record shows very high values before 18 kyr BP and a decreasing trend towards lower DCF values until 13 kyr BP. The following period between 13 and 11 kyr BP is characterized by higher, and highly variable





DCF values of between 40 and 50 % and exhibit rapid fluctuations on decadal time scales as seen in a 9 % drop in a time
interval of only 60 yr modeled age at approximately 11.9 kyr BP.

### 4.3 Mg/Ca Ratio

Mg/Ca ratios range from 0.01100 ± 0.00033 to 0.03217 ± 0.00097 and display a continuous decrease from 22 to 12 kyr BP.
After this decrease, Mg/Ca ratios rise sharply to their highest values around D1. After a few very high values, the ratio drops
back to the level measured before the major excursion (**Fig. 5**).

## 5 Discussion

### 5.1 U-series Dating and Age-Depth Model

The revised chronology for stalagmite M1-5 is far more accurate and precise in comparison to the previously published
chronology which was based on much fewer U-series ages (Shakun et al., 2007). Average uncertainties of the corrected U-
series ages have decreased from 1.19 ± 0.30 % in Shakun et al., 2007 to 0.408 ± 0.161 % in this study. The refined age record
helped to identify the two growth discontinuities (see **Fig. 3**). $^{238}$U concentrations remain fairly constant around 2 ppm before
14 kyr BP when a dilution effect is visible and concentrations decrease to less than 1 ppm ~11 kyr BP. The StalAge model
reveals a general trend towards higher growth rates in the younger part of the stalagmite with highest growth rates of 0.30 mm
yr$^{-1}$ at around 13 kyr BP, more than a factor 2 higher than during the last glacial period. The general growth patterns can even
be distinguished by mere optical examination of the stalagmite lamination. At a depth of ~1.5 m (17 kyr BP) a transition from
marked lamination and darker colors to lighter layer coloring is visible. Synchronously, the width of the stalagmite decreases
gradually from more than 15 cm to less than 10 cm in the younger half (see **Fig. 2**), where the growth rate is evidently highest.
According to stalagmite growth modeling studies (Dreybrodt, 1999; Kaufmann and Dreybrodt, 2004) higher drip rates and
higher temperatures cause a larger width of stalagmites. Nevertheless, precipitation has increased over the growth of stalagmite
M1-5 which will be further discussed in section 5.3. Since this probably caused the drip rate to rise in the cave, our study
contrasts the findings of the aforementioned studies on stalagmite growth. Shakun et al., 2007 postulated a temperature increase
on Socotra Island of about 2–3.5 °C from the LGM to the Holocene. Therefore the changes in stalagmite width might in this
case be caused by temperature increase rather than higher drip rates.

### 5.2 Reservoir Ages and Implications for Radiocarbon Calibration

With the implemented age model for stalagmite M1-5, $^{14}$C reservoir ages can be accurately assessed for the time interval of
stalagmite growth overlapping with the dendrochronological records till 12.4 kyr BP and the wiggle-matched floating Late
Glacial pine tree-ring chronology back to 14 kyr BP (Hua et al., 2009; Schaub et al., 2008). Both are a substantial part of the
intercalibration record IntCal13 (Reimer et al., 2013). For this time period the atmospheric $^{14}$C concentration is well known.
**Figure 6** compares the record of M1-5 to the speleothem based $^{14}$C records previously implemented in IntCal13 (Reimer et



al., 2013). For the time interval overlapping with the tree-ring based atmospheric $^{14}$C calibration data (including floating tree

ring chronologies) stalagmite M1-5 reveals high reservoir age variability over centennial and decadal time scales, equivalent

to DCF values between 27 % at approximately 13 kyr BP and 50 % at approximately 11.9 kyr BP. A doubling of DCF values

is observed at the onset of the YD around 12.9 kyr BP (**Fig. 6**). Furthermore, fast increases and decreases by a few percent

DCF occur frequently between 12.9 and 11.9 kyr BP. The data implicating these rapid fluctuations have been replicated several

times to exclude the possibility of outliers. This highlights the large reservoir age variability of stalagmite M1-5 on sub-

centennial time scales which puts this $^{14}$C record in stark contrast to the IntCal13 stalagmite records.

For instance, no comparable variability is visible in the Hulu cave speleothem H82 where the DCF remains rather constant at

5.6 ± 0.8 % (Southon et al., 2012). The Bahamas stalagmite records show substantial and systematic changes in DCF between

11.2 and 12.7 kyr BP (Hoffmann et al., 2010) which coincides with the YD cold reversal and was therefore attributed by the

authors to vast changes in local vegetation and climatic conditions. Notwithstanding these effects and the resulting large

uncertainty in the corrected atmospheric $^{14}$C concentrations the Bahamas record was included into the IntCal13 data, as the

benchmark Hulu cave record was not available. The $^{14}$C record of stalagmite M1-5 does not predominantly reflect atmospheric

$^{14}$C variations due to the large offset to the atmosphere, and the high variability across the tree-ring based period. Hence, M1-

5 cannot be used as a contribution for atmospheric $^{14}$C calibration. This finding is in contrast to the Hulu cave speleothems

which come from an a priori comparable temperate setting which is under the influence of pronounced monsoonal patterns

(Southon et al., 2012).

Potential atmospheric $^{14}$C variations are clearly overprinted in stalagmite M1-5 by sub-centennial variations of the reservoir

ages, which must be driven by site-specific factors such as changes of soil carbon age, open–closed system conditions

(Fohlmeister et al., 2011; Griffiths et al., 2012) or short-term developments affecting soil carbon dynamics above Moomi Cave.

Other potentially relevant mechanisms include the influence of non-carbonic acids such as sulfuric acid (Bajo et al., 2017).

Since most of the aforementioned factors are influenced by climate, stalagmite M1-5 provides an excellent opportunity to

study climatic controls on the DCF values in stalagmites.

### 5.3 Climatic controls of DCF at Moomi Cave

Low DCF values in stalagmites are primarily caused by open system recharge conditions, with enhanced uptake of soil $CO_2$,

and limited water-bedrock interaction. Both conditions prevail at Hulu cave, where the seepage water percolates through

predominantly sandstone rather than carbonate limestone (Cheng et al., 2018). In our study very high average DCF values of

56.2 ± 3.4 % are revealed during the last glacial period between 18 and 27 kyr BP with a maximum of 65 % approximately

20.6 kyr BP (**Fig. 7a**). Such high DCF values are only conceivable under closed system limestone dissolution conditions

(Hendy, 1971). However, in the conventional approach considering only the two carbon end-members from near-atmospheric

soil gas $CO_2$ and the $^{14}$C-free carbonate from dissolved limestone, DCF can reach maximum values of 50 % (Fohlmeister et

al., 2011; Griffiths et al., 2012; Hendy, 1971). DCF higher than 50 % have been recently reported in a speleothem from Corchia

Cave (Italy) as a suggested consequence of additional dead carbon input by enhanced limestone carbonate contribution to DIC





through the presence of other organic or non-organic (sulfuric) acids in the soil zone (Bajo et al., 2017). This process also influences stable carbon isotopes in the stalagmite calcite, i.e. $\delta^{13}C$, which shift towards the values observed in the limestone host rock. However, overall $\delta^{13}C$ in M1-5 (**Fig. 7c**) range mostly between -4 and -8 permil (overall average -6.2 ± 1.4 permil)

with corresponding DCF values varying vastly from 27 to 65 %. Furthermore, no significant correlation between $\delta^{13}C$ and DCF can be observed ($r^2 = 0.21$) in M1-5. It is therefore unlikely that increased limestone dissolution by sulfuric acids is the key factor for DCF variations in stalagmite M1-5. Alternatively, aged SOM could cause the observed enhanced DCF values (Genty et al., 2001). Fresh labile soil organic matter as well as older carbon stocks in deeper soil layers on the Moomi Plateau may have contributed to soil gas $CO_2$ creating $^{14}C$-depleted $CO_2$ (Fohlmeister et al., 2011; Trumbore, 2009) which, together

with nearly closed system karst dissolution conditions, can cause DCF higher than 50 %. For instance, if an extreme case base line DCF of 50 % under closed system conditions is assumed for the record previous to 18 kyr BP, an age of the active carbon pool of 2500-3000 yr can generate DCF values between 63 and 65 %. If the carbon pool is composed of one young (root respiration) and one old carbon reservoir (microbial decomposition of aged SOM) in equal proportions, the aged carbon contributing to soil $CO_2$ must be older than 5000 yr and contribute a share of 50 % to the soil gas budget in order to cause such

extreme DCF values. None of these scenarios have been reported before in a study on speleothem $^{14}C$ which makes M1-5 a unique record in terms of aged SOM contribution to stalagmite reservoir age.

All available geochemical proxies (shown in **Fig. 7**) including DCF (**7a**) reflect a clear climate signal: The LGM is manifested in this stalagmite by low net-infiltration levels and rather low precipitation on the archipelago (Shakun et al., 2007). In a recent study by DiNezio et al., 2018 it was argued that the exposure of the Sahul shelf during the Last Glacial Maximum created a

positive ocean-atmosphere feedback-loop causing a drier/wetter dipole across the Indian Ocean and subsequent hydroclimate changes, i.e. reduced moisture levels in the Arabian Sea. The M1-5 multi-proxy record suggests that towards the end of the glacial period the regional climatic conditions on Socotra Island likely shifted to a higher net-infiltration which is reflected in a systematic long-term decrease of the Mg/Ca ratio from approximately 20–15 kyr BP. We interpret the Mg/Ca decrease here as a proxy for incongruent host rock dissolution and prior calcite precipitation effect reflecting shifts from drier to wetter

conditions above the cave (Fairchild et al., 2000), while peaks in Mg/Ca coincide with the two growth discontinuities and thus dry periods. This interpretation is supported by numerous studies that showed a negative correlation between Mg/Ca ratio in stalagmite calcite and precipitation amount (Fairchild and Treble, 2009; Flohr et al., 2017; Noronha et al., 2014; Warken et al., 2018).

$\delta^{234}U_{init}$ can also be used to derive changes in infiltration and thus drip rate (Dreybrodt, 1999). In our study, the increasing but

still negative values in $\delta^{234}U_{init}$ (**Fig. 7e**) after 22 kyr BP point to a shift towards less leached host rocks or a stronger contribution of excess $^{234}U$ from the leached rocks. This change coincides with a doubling of the growth rate from slightly below 0.10 mm yr$^{-1}$ to more than 0.20 mm yr$^{-1}$, suggesting that $\delta^{234}U_{init}$ is in fact influenced by infiltration changes. It may also be affected by the geochemical composition of the seepage water (Zhou et al., 2005). However, in comparison to the other proxies for this record, it appears most likely that the drip rate in Moomi Cave has increased due to higher net-infiltration. Our

records therefore contrasts the findings of Dreybrodt, 1999 and Kaufmann and Dreybrodt, 2004 where higher drip rate is



correlated to increased stalagmite width. Stalagmite M1-5 on the other hand decreases in width (see **Fig. 2**) towards higher infiltration in a time where temperatures have increased by 2–3.5 °C and precipitation rises (Shakun et al., 2007). This climatic shift is initiated during peak glacial conditions 20 kyr BP and is also reflected in higher seepage water excess $^{234}$U, marked through the vanishing depletion of $^{234}$U (-25 permil after 16 kyr BP).

The inferred increase in precipitation over Socotra coincides with the steady warming of the southern hemisphere until 15 kyr BP, although it likely exhibits a yet unresolved secular variability. Synchronously $\delta^{13}$C decreases by 2‰ and resolves secular sub-centennial variability of the speleothem calcite composition. During the time in which northern hemispheric climate goes through swings between the warm and humid B/A and cold and dry YD (Ivanochko et al., 2005; Schulz et al., 1998), $\delta^{13}$C and Mg/Ca in M1-5 remain at their respective lowest values. In contrast to $\delta^{13}$C values, the $\delta^{18}$O values show a moderate decrease

by 1.5‰ across Termination I (20–11 kyr BP), which essentially occurs rather suddenly at ~14.8–15 kyr BP. Stalagmite M1-5 $\delta^{18}$O values are mainly interpreted in terms of the so-called "amount effect" describing the negative correlation between rainfall amount and $\delta^{18}$O (Dykoski et al., 2005; Griffiths et al., 2010). Shakun et al., 2007 have used this relation to infer a rise in rainfall towards the end of the last glacial, especially after 15 kyr BP. Both stable isotopes thus share a common deglacial climate trend with punctuated strong correlation during growth perturbations (dry events), but differ significantly through the

period from 19–14 kyr BP, indicating a dominant influence of vegetation above the cave which affected $\delta^{13}$C, but not $\delta^{18}$O. Although all available proxies point to an increase in precipitation after the LGM, M1-5 $^{14}$C data does not reflect the processes predicted if the established dependence of calcite dissolution systematics on soil infiltration holds. In contrast to studies from temperate zones (Genty et al., 2001; Griffiths et al., 2012; Lechleitner et al., 2016; Noronha et al., 2014), the predicted positive feedback of rainfall amount on DCF does not appear to apply here. Instead, increasing rainfall and infiltration respectively

seems to go along with a shift to a less closed carbonate dissolution system and an increased forcing towards higher $^{14}$C levels in the soil gas $CO_2$ by enhanced root respiration compared to a decreasing contribution by $CO_2$, which originates from decomposition of aged SOM. This decoupling of precipitation and DCF is evidence for a climate-induced forcing on stalagmite $^{14}$C which is at least to some extent independent of the direct influence of soil humidity on calcite dissolution. A proposed scenario for M1-5 includes that – in contrast to temperate, more humid settings – vegetation in (semi-)arid regions like Socotra

Island, where precipitation is sparse and occurs only seasonally when the ITCZ crosses the archipelago, is much more sensitive to small changes in net-infiltration. Previous studies showed that an increased monsoonal activity during Termination I (Gupta et al., 2003; Overpeck et al., 1996; Shakun et al., 2007), can have dramatic effects on the prevalence and spread of plants in arid and semi-arid areas (Lotsch et al., 2003). Hence, for this study an increase in soil $CO_2$ input by active vegetation caused by increasing net-infiltration or precipitation at the end of the last glacial period is the most likely cause for the observed soil

carbon dynamics, ultimately causing strongly decreasing DCF.
Since the ITCZ passes Socotra Island twice a year causing increased precipitation, the question arises whether the observed trends are linked to a shift in the seasonal patterns of either summer or winter monsoon season or the intermediate ITCZ migration. Overall, we can only speculate on the role of the monsoon in shifting the local climate from a glacial possibly



southern hemisphere driven state with a stable state aged soil carbon pool (summer monsoon driven) to a northern (or mixed)

hemisphere state beyond 15.5 kyr BP with an increasingly labile, non-steady state soil carbon age profile.

Consequently, we have traced a clear multi-millennial increase in precipitation on the archipelago from the LGM to the 15 kyr northern hemisphere sudden warming, which is accompanied by major changes in soil activity and thus vegetation.

**5.4 Glacial Termination Climate Dynamics on Socotra Island**

Beyond the strong B/A warming the tracers resolve sub-centennial variability of DCF, $\delta^{234}U_{init}$, and Mg/Ca. This is possibly

indicative of a strong local climate-monsoon coupling that is only resolved in the area with the highest temporal data density in the record. Nonetheless, one striking feature stands out: At approximately 13 kyr BP, DCF abruptly jumps to values higher than 40 % indicating sudden changes in soil carbon cycling. It was argued that sudden events caused by climatic variations or wildfires can have a major influence on soil carbon dynamics (Treble et al., 2016; Trumbore, 2009). For instance an event like a wildfire or landslide could have drastically changed the soil configuration by depleting active vegetation and thus halting

$CO_2$ input at near-atmospheric $^{14}C/^{12}C$ ratio (Coleborn et al., 2016; Markowska et al., 2019), and could therefore account for the sudden increase in DCF. Except for short-term excursions, all other proxies remain on average at virtually the same level over the event with Mg/Ca ratios indicating persistent high precipitation during the YD. Hence, this feature of the M1-5 record is only visible in the $^{14}C$ data.

The high data resolution (exemplified by DCF and Mg/Ca records in **Fig. 8**) reveals marked variability in DCF on sub-

centennial timescales, exemplified by a sudden decrease of almost 10 % within less than 100 yr. Several of these peaks can be seen throughout the whole period of high temporal resolution, although they are not represented in Mg/Ca variations and correlation in this period is insignificant (r²=0.23). The high volatility in $^{14}C$ reservoir suggest a pronounced influence of local effects on the soil carbon dynamics on Moomi plateau. Other proxies, while still showing some secular variability, do not suggest a persistent change for the YD. $\delta^{13}C$ values remain at comparable levels at the end of the glacial period which indicates

a high influence of organic or biogenic $CO_2$ on stalagmite formation and contradicts enhanced limestone carbonate dissolution. $\delta^{18}O$ values remain roughly within the state which suggest intensified precipitation during B/A which was attributed to a stronger monsoon (Shakun et al., 2007). Hence, a persistent change in climatic conditions on Socotra Island appears not likely for the YD. Although from our data we cannot conclusively derive the direct cause for the observed fluctuations in DCF which are not represented in hydrological proxies, we propose that labile conditions of the soil carbon pool above the cave or short-

term changes in carbonate dissolution systematics might have played a vital role.

**6 Conclusions**

Stalagmite M1-5 from Moomi Cave, Socotra Island, is by many measures similar to those used in atmospheric $^{14}C$ calibration. Precise U-series dating revealed an overall growth rate of more than 0.13 mm yr$^{-1}$. The geographical setting in a subtropical climate in Western Arabian Sea under the influence of the East African–Indian monsoon is comparable to those of the Bahamas

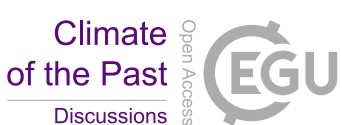

and Hulu Cave speleothems (Beck et al., 2001; Hoffmann et al., 2010; Southon et al., 2012) that have contributed to [14]C calibration. However, the DCF values of M1-5 are very high, among the highest that have been reported and additionally show distinct trends over the entire growth period of the stalagmite, thus proving the climatic impact on speleothem [14]C. Large short-term DCF changes on sub-centennial time scales emphasize the importance of local carbon cycling variability on stalagmite [14]C. Whereas the Hulu Cave record (Cheng et al., 2018; Southon et al., 2012) with its low DCF, that remains constant even over climate changes, has been referred to as the "Holy Grail of [14]C dating", records like M1-5 are crucial for the understanding of soil carbon cycling and karst hydro-geochemistry.

We conclude that M1-5 grew under near-closed conditions with a high influence of aged SOM prevailing throughout the LGM. This causes the DCF to exceed its theoretical maximum of 50 % before increasing precipitation and wetter conditions triggered a gradual increase in [14]C enriched $CO_2$ input, most likely caused by denser vegetation coverage on the plateau above the cave. With most proxies pointing to a higher net-infiltration rate towards Termination I, M1-5 shows a trend opposed to previous observations suggesting a positive correlation between soil humidity and DCF due to the influence of hydrology on carbonate dissolution. Rather, the data reveals a probable intensified vegetation occurrence on Moomi plateau causing a higher influx of $CO_2$ at high [14]C levels into the soil gas regime that shifts DCF to lower levels. These findings motivate the implementation of [14]C in stalagmites as a tracer for vegetation and emphasize the distinction between temperate and humid settings and semi-arid or arid regions when assessing the influence of precipitation changes on DCF in stalagmites. Soil carbon dynamics and the influence of vegetation and SOM seem to exert significant forcings on DCF, exemplified in our record by both the long-term trend towards Termination I and the vast variability on secular time scales during YD.

Our work is an important contribution to future efforts to understand soil dynamics and their connection to stalagmite reservoir ages as well as additional efforts in the search for adequate [14]C calibration records.

**Data Availability**

The data presented in this paper were uploaded to the PANGAEA data library (https://doi.org/10.1594/PANGAEA.906003) and will be available after acceptance of this paper.

**Author Contributions**

The scientific project was designed and conducted by JF and NF. ST performed the sampling for all measurements and the data evaluation and wrote the paper with contributions by the other co-authors. JA performed the U-series measurements and quality assessment. AM and SJB played a major role in the investigations and expedition that led to the collection of the stalagmite. RF conducted the [14]C measurements. ASR performed the elemental measurements. All co-authors contributed to the discussion of the results and interpretations.





**Conflict of Interest**

The authors declare that they have no conflict of interest.

**Acknowledgements**

This study was funded by the DFG Research Project Grant FR1341/3 and the DFG large equipment grant INST35-1143-1
FUGG. We thank René Eichstädter at the Institute of Environmental Physics for conducting the MC ICP-MS measurements
with the help of Sandra Rybakiewicz, Hanna Rosenthal and Carla Roesch as well as Marleen Lausecker for the ICPQMS
measurements. We also thank Silvia Rheinberger and Christian Scholz for the elemental measurements at the Institute of Earth
Sciences at Heidelberg University.

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





**Figures**



**Figure 1:** uppermost figures show the geographic position of the Socotra archipelago (red circle) in the Western Arabian Sea, east of the coast of Somalia with the prevailing monsoonal surface wind patterns in boreal summer (a) and boreal winter (b) relative to the ITCZ (graphic adapted from Fleitmann et al., 2007). In a topographic map (c), the position of Moomi Cave in the East of Socotra Island on the Moomi Plateau is indicated by a white star.




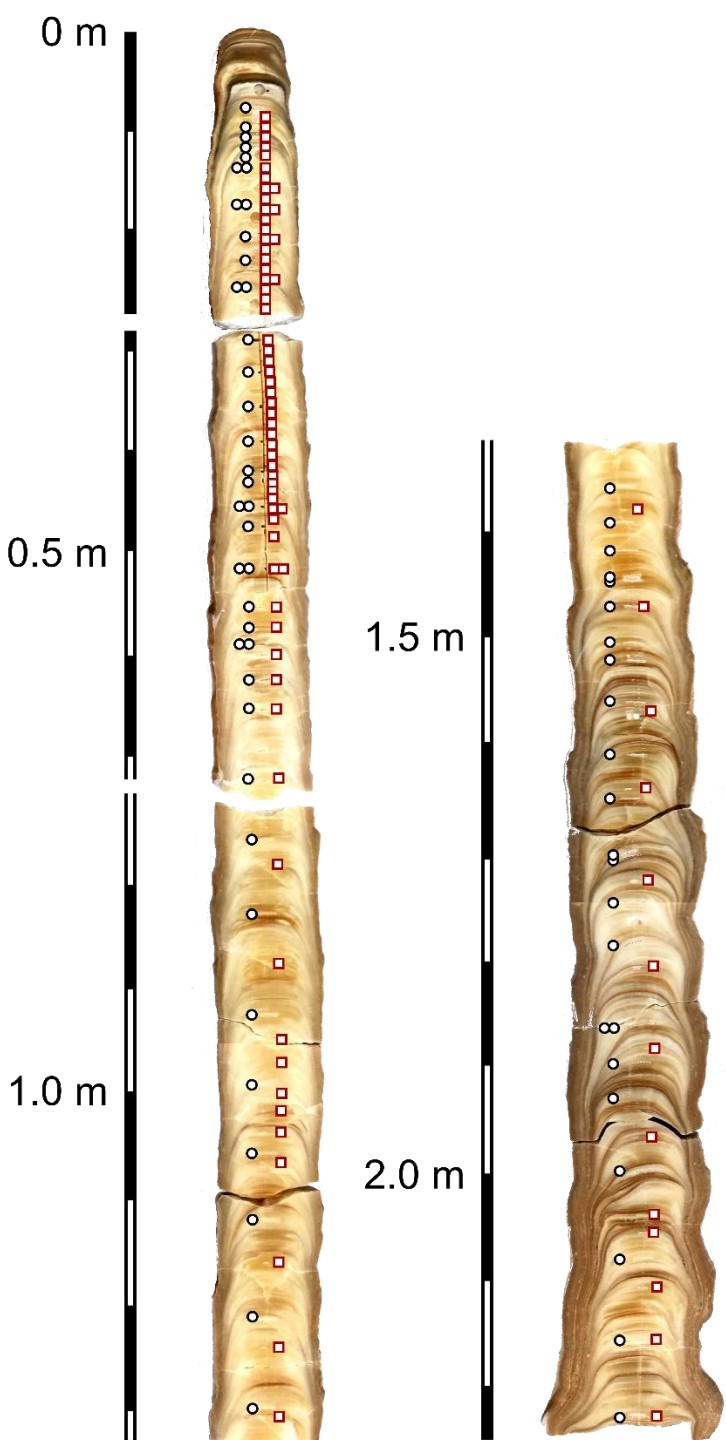

**Figure 2: Composite photograph of stalagmite M1-5 from Moomi Cave. Black circles indicate where U-series dating samples were taken, red squares represent $^{14}$C sample locations.**



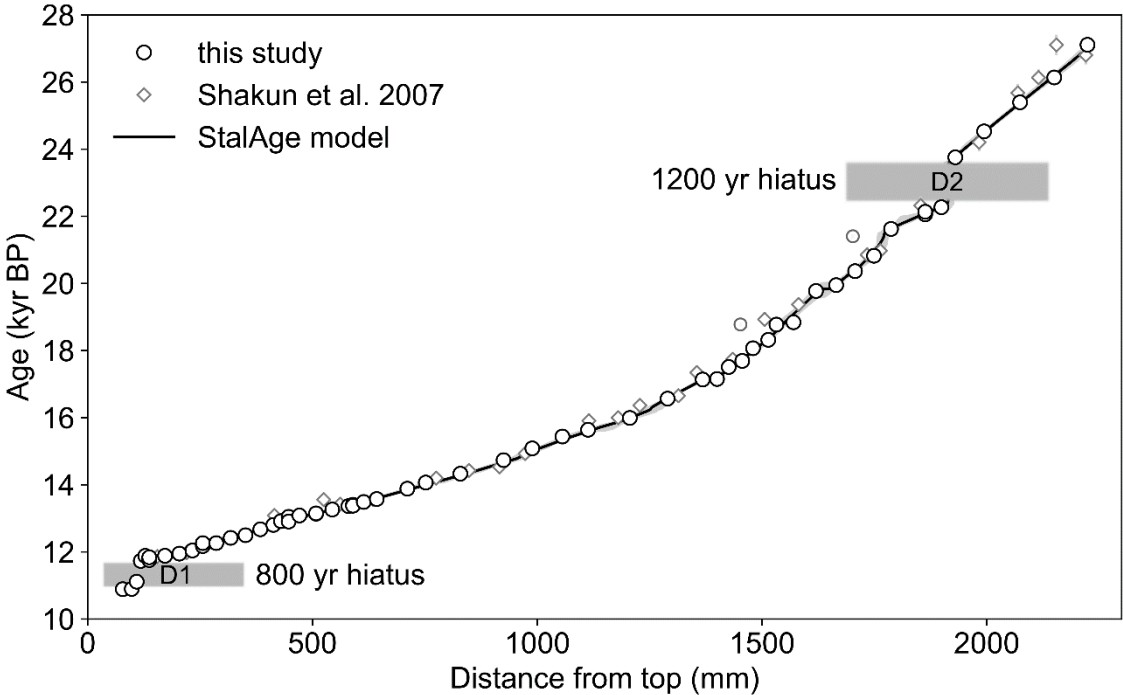

**Figure 3: U-series ages vs. depth. Error bars reflect 2σ statistical uncertainty but are mostly covered by the data points. The solid line represents the established age model obtained by sectional implementation of the StalAge algorithm** (Scholz and Hoffmann, 2011)**. Two data points implying age inversions (smaller circles) in an otherwise steady record were omitted. Diamonds represent data from a previous study** (Shakun et al., 2007)**. Growth discontinuities are highlighted by grey bars.**


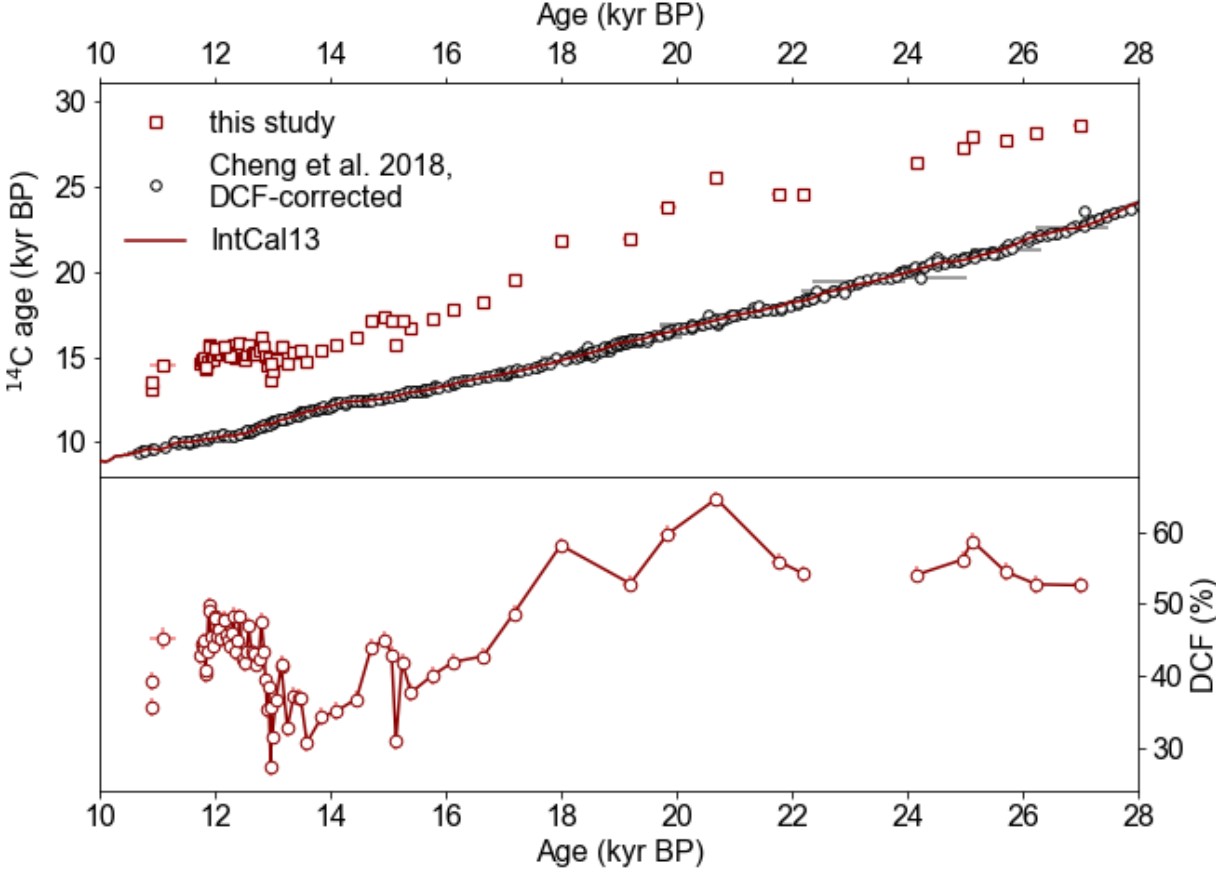

**Figure 4: Upper graph (a) shows the $^{14}$C ages of stalagmite M1-5 (red squares) plotted vs. age derived from the StalAge model. Error bars are mostly smaller than sign size. Also shown is the atmospheric $^{14}$C age (solid red line) from** Reimer et al., 2013**, with the recently published DCF-corrected stalagmite record from** Cheng et al., 2018**. The DCF of stalagmite M1-5 is shown in the bottom figure (b).**

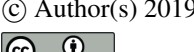



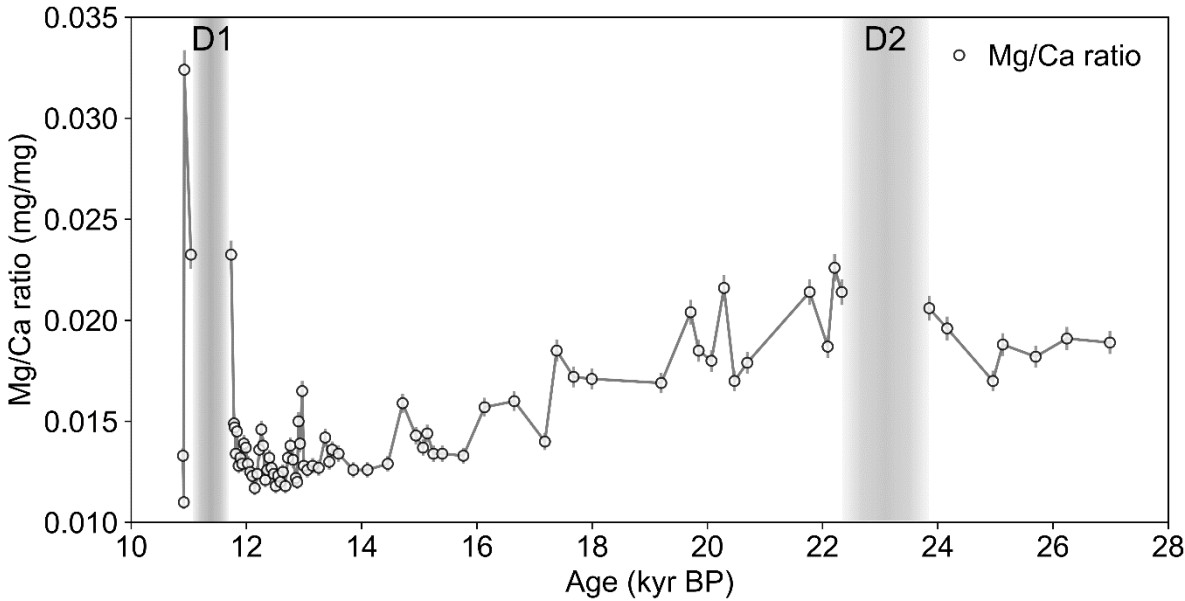

**Figure 5: Mg/Ca ratio of stalagmite carbonate vs. modeled age. Error bars reflect the data reproducibility (overall 3 % of measured values). The data reflects a general decreasing trend from the LGM towards Termination I which is interrupted by highly increased values around D1 at approximately 11.7 kyr BP. Both hiatuses inferred from U-series dating are shaded in grey.**




**Figure 6: Very high and varying DCF for stalagmite M1-5 (this study) in comparison to records included in calibration data sets** (Beck et al., 2001; Hoffmann et al., 2010; Southon et al., 2012)**. Whereas H82 DCF values are low and constant, the Bahamas DCF are higher and show non-random structures between 11.2 and 12.7 kyr BP. Empty signs indicate the used reference area for the respective calibration study, the shaded colored areas represent mean values and standard deviation of DCF values in the respective**
**reference areas. Areas of available tree ring records** (Reimer et al., 2013) **are indicated on top of the graph.**



**Figure 7: Combined results from stalagmite M1-5. DCF (a), growth rate from StalAge modelling (b), adapted stable isotope data from** Shakun et al., 2007 **(c), Mg/Ca mass ratio (d) and δ²³⁴U from this study and** Shakun et al., 2007 **(e). The grey bars indicate Bølling-Allerød interstadial (B/A) and Younger Dryas (YD) cold reversal. Oxygen data were interpreted as an increase in monsoonal rainfall over Termination I. This is corroborated by Mg/Ca precipitation proxy data, showing its lowest values in the YD, where growth rate is highest, with a decreasing trend over the end of the last glacial period. DCF shows an obvious decreasing trend after the LGM until it increases at the onset of YD. δ¹⁸O reaches its lowest values during the B/A warm period, indicating an intensified monsoonal precipitation** (Shakun et al., 2007)**.**

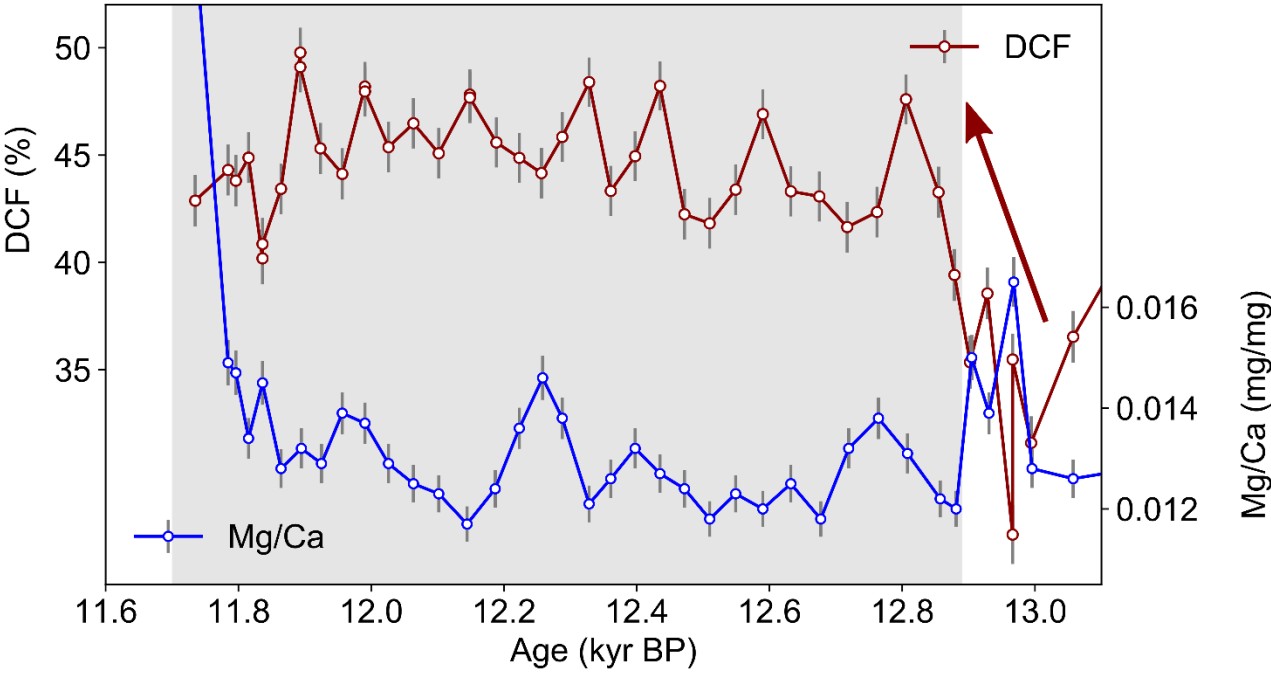

**Figure 8: DCF and Mg/Ca of stalagmite M1-5 for the YD. DCF increases sharply at the onset of the YD (red arrow). The high data resolution reveals pronounced changes in DCF on extremely short time periods, highlighting the vast soil dynamics during the YD (grey area) where growth rate is the very high (>0.20 mm yr$^{-1}$). The hydrological proxy Mg/Ca remains relatively low, indicating a high infiltration rate. The correlation between DCF and Mg/Ca is insignificant in this time (r$^2$=0.23).**