# Peer review of "Climate Induced Speleothem Radiocarbon Variability on Socotra Island from the Last Glacial Maximum to the Younger Dryas"

_Climate of the Past, 2019_

## Referee Comment (RC1) · Anonymous Referee #1 · 8 Oct 2019

General comments: Therre and co-authors present a new 14C record from Socotra Island, spanning the last deglaciation. They find very high and variable DCF values during the late glacial, which can only be explained by a combination of closed system dissolution regime and a high contribution of pre-aged soil organic matter. Through a multi-proxy approach, the authors can interpret changes in DCF with respect to synchronously occurring changes in infiltration and rainfall amount. This has important implications for the use of stalagmites for extending the 14C calibration curve, which as this and other studies show must occur on a case by case basis and is highly dependent on local climate and soil/vegetation dynamics. On the other hand, records such as this can provide novel and highly timely insights into soil carbon dynamics over time,

and their sensitivity to climatic change.

I enjoyed reading this paper, which I think is well within the scope of Climate of the Past. The scientific approach and methods are valid and outlined in great detail. The results are discussed appropriately, considering previous related research. The quality of the paper is already high, it is overall well written, with clear and concise presentation of results and conclusion and nice figures. I therefore recommend publication after minor revisions.

Specific comments: Page 7, lines 205-207: I'm not sure I follow the conclusions of the authors that temperature drives the stalagmite width change. If higher drip rates and higher temperatures cause a larger width of stalagmites, then how can it be explained that the width of the stalagmite decreased when temperatures increased after the glacial? There might be a typo in the above sentence explaining the mechanism...

Page 9, discussion of causes for high DCF values. I agree with the authors that the proposed scenario (closed system dissolution and an old SOM pool contributing carbon on top of that) is likely the only process that can explain the observed DCF values in this case, but I would like to see a bit more discussion of the soil/karst system at Moomi cave. I assume that there have not been any investigations on the soil, which is unfortunate. However, it would still be nice to see a couple of sentences discussing how likely the presence of pre-aged organic matter is in the soil or karst above the cave (how thick is the soil and what type of soil is it, what kind of vegetation covers the cave, is there any chance that there is a reservoir of ground air that could contribute pre-aged $CO_2$ to the drip water?).

Page 10, discussion of d18O and d13C signal over glacial termination. The authors suggest that the slight differences between d18O and d13C during the deglaciation, in particular the gradual decrease in d13C and more sudden termination in d18O are related to d18O registering the amount of precipitation (amount effect), while d13C is affected by vegetation changes. While this is one possibility to explain these different

trends, it strikes me that the long-term trend in d13C is also reflected in the other hydrology proxies, in particular in Mg/Ca. Is it possible that d18O and d13C reflect a decoupling of regional moisture dynamics and local infiltration, as suggested for other Asian monsoon records (e.g., Myers et al., 2015, GRL, Cheng et al., 2016, Sci. Rep.)?

Technical comments: Page 6, lines 172-173 (14C results): I had to read this sentence several times, as to me it reads as if the 14C ages of the stalagmite were younger than the atmosphere. Would it be possible to rephrase to "the 14C ages of the stalagmite are significantly older than..."?

Page 10, line 298: I find the references to the Griffiths 2012 and Lechleitner 2016 papers here out of place, as these are both from tropical cave systems and therefore not from temperate environments. Maybe it would be worth rephrasing to "more humid" compared to Socotra?

Page 11, line 332: Would it be possible to provide a p value as well for the correlation between DCF and Mg/Ca? Also, note that in the text the authors refer to r2, while in the figure caption for figure 8 they use r.

Figure 3: It's quite hard to distinguish the ages from this study from the previous Shakun ages. Maybe the new ages could be highlighted by colouring the inside of the circles in grey? This would also help distinguish them from the omitted ages.

Figure 5: Please explain in the caption what D1 and D2 stand for.

Figure 6: "treerings" on the top of the figure should be "tree rings"

Figure 7: What do the different symbols for d234U stand for? Please explain in the caption or with a legend in the figure.

Figure 8: I would move the description of the YD (grey area) to the end of the first sentence in the caption, because as it is now it reads a bit confusingly (it seems like the grey box says something about soil dynamics).

[Figure]

---

## Referee Comment (RC2) · Anonymous Referee #2 · 10 Jan 2020

The manuscript of Therre et al. presents a new study where radiocarbon and U-Th dates were combined to reveal DCF variability in a speleothem from a Moomi cave on Socotra Island. The record covers last glacial to early Holocene time period with two well defined hiatuses. Very high DCF values during the last glacial are interpreted as a consequence of significant contribution of aged soil organic matter and almost completely closed system dissolution conditions. DCF record exhibits overall decrease from last glacial to the onset of Younger Dryas event. To explain this long-term decreasing trend as well as relatively large variability DCF results were paired with other speleothem proxies (growth rate, Mg/Ca, d18O, d13C and d234Uinit) from the same specimen and interpreted in terms of past climate changes and its influence on carbon

soil dynamics at this site. The manuscript is well organized. The introduction, methodology and results sections provide enough information and are mostly well written. The results are clearly presented and well discussed leading to conclusions worth of publication. There is however some space for improvement in clarity of the text and I would suggest that a native English speaker improves it before it is accepted for publication. A few line by line comments are provided below.

Throughout the manuscript: I find using a term "reservoir age" together with "DCF" confusing. At places you consider them as synonyms at others not. Either make a clear distinction between the two terms or stick to the one. DCF is more often used in speleothem-based studies so I would advise choosing it over reservoir age for clarity.

Lines 59-60 Please consider rewriting this sentence. As stated above the whole manuscript would benefit if a native English speaker improves it.

Line 164 Which part of the procedure was replicated? All the steps including subsampling, or only 14C measurements? Please clarify.

Line 168 Please replace word ËİabsoluteËİ with U-Th modelled.

Line 196 Add word ËİatËİ or ËİsinceËİ before ∼ 11 kyr BP.

Line 198 Add word ËİofËİ between factor and 2.

Lines 200-207 I find discussion related to the changes in the width of the stalagmite irrelevant in this manuscript and hard to follow. Please consider removing it from the manuscript.

Line 286 What do you mean by secular variability? Please rewrite here and later in the text.

Line 298 Please remove citations of Griffiths et al. and Lechleitner et al. here as the cave sites in these two studies are not located in temperate climate zones.

---

## Author Comment (AC1) · 17 Jan 2020

Dear Climate of the Past Editorial Board, Dear reviewers

All the authors would like to thank the two anonymous reviewers for their time spent on reading and evaluating the manuscript, and for their constructive comments on our study. The suggested changes and additions have significantly improved the quality of our manuscript and are highly appreciated by all contributors (see Acknowledgements).

Below, we have listed our responses to all comments and the changes made in the manuscript.

**Response to Reviewer #1**

1. Page 7, lines 205-207: "I'm not sure I follow the conclusions of the authors that temperature drives the stalagmite width change. […]"

   This paragraph was corrected from line 202 to the end of the paragraph to be in accordance with the studies by Dreybrodt and Kaufmann. Increased stalagmite diameter and growth rate are both consequences of higher temperatures and increased drip rate. However, for a time with most likely higher drip rates due to increased precipitation a higher growth rate is observed but stalagmite diameter has decreased. This is in conflict with the studies mentioned above, which is what this paragraph now highlights.

2. Page 9, discussion of causes for high DCF values: "[…] I would like to see a bit more discussion of the soil/karst system at Moomi cave. I assume that there have not been any investigations on the soil, which is unfortunate. However, it would still be nice to see a couple of sentences discussing how likely the presence of pre-aged organic matter is in the soil or karst above the cave (how thick is the soil and what type of soil is it, what kind of vegetation covers the cave, is there any chance that there is a reservoir of ground air that could contribute pre-aged CO2 to the drip water?)"

   The reviewer is correct in assuming that there are no studies (to the knowledge of the authors) on the particular conditions of the soil zone on the Moomi plateau above the cave. The presence of aged soil organic matter is however very likely considering several assessments of the current local vegetation situation and flora (Mies and Beyhl, 1996; Popov, 2008). In these studies it is elaborated that on the Socotran limestone plateaus and particularly in the area of Moomi cave there is annual vegetation like grass as well as permanent growth of shrubs, especially in sheltered valleys, where the soil is well developed and dense thickets occur. The presence of the endemic tree species Dracaena cinnabari is also suggested by the aforementioned studies for the region. The landscape is described as rocky limestones where pockets of dark rich soil can form (Popov, 2008) giving rise to stabilizing carbon pools. Under these circumstances, soil matter can aggregate and decay over long periods of time, thus contributing to the soil $CO_2$ reservoir. These aspects were worked into the discussion with four additional sentences starting on page 9 in line 254 after the references.

3. Page 10, discussion of d18O and d13C signal over glacial termination: "[…] it strikes me that the long-term trend in d13C is also reflected in the other hydrology proxies, in particular in Mg/Ca. Is it possible that d18O and d13C reflect a decoupling of regional moisture dynamics and local infiltration, as suggested for other Asian monsoon records (e.g., Myers et al., 2015, GRL, Cheng et al., 2016, Sci. Rep.)?"

   It is true that the observed changes on Socotra Island after the LGM do not necessarily require an increase in total precipitation amount, but rather a shift towards wetter soil conditions, as described by Cheng et al., 2016, Sci.Rep.. In our study, wetter soil conditions are corroborated by lower Mg/Ca and $\delta^{13}C$ values after the LGM whereas $\delta^{18}O$ is decoupled from this trend: it stays

fairly constant. In the study from Myers et al., 2015, GRL, similar observations were made in north-eastern Indian stalagmites where $\delta^{18}O$ variations were attributed to reflect changes in monsoonal moisture transport routes or sources while precipitation amounts remained rather constant. This exemplifies a decoupling of precipitation amount changes and $\delta^{18}O$ variation in stalagmite calcite. For Socotra Island, wetter soil conditions are not necessarily a result of increased precipitation amounts, which would be observable by lower $\delta^{18}O$, but might as well be caused by shifts to a more distributed annual rainfall pattern. In this case, higher net-infiltration levels in the soil on Moomi plateau create favorable conditions for more vegetation growth, resulting in both lower Mg/Ca and $\delta^{13}C$ and a decreasing DCF in stalagmite calcite. We have added these crucial aspects to the manuscript on page 10 after the sentence in line 295.

Minor comments:

- Page 6, lines 172-173 (14C results):
  The sentence was rephrased by replacing "higher" by "older" to clarify the radiocarbon age terminology.

- Page 10, line 298:
  The references to the Griffiths 2012 and Lechleitner 2016 papers were kept in the manuscript, although the two cave sites in those studies are from tropical cave systems. To maintain coherence the sentence was rephrased to read "more humid" instead of "temperate".

- Page 11, line 332:
  The p-value of the discussed correlations were added to the correlation coefficients, where they occur. The manuscript was also checked for coherence concerning the use of r² and r.

- Figure 3:
  The data of this study used for the age modelling were highlighted in the graph by colouring the inside of the circles in grey to provide a better contrast to both the omitted ages and the previous results by Shakun et al., 2007.

- Figure 5:
  Explanations of D1 and D2 were added to the caption to match the explanations in the text.

- Figure 6:
  The typo was corrected to read "tree rings" in the figure, as well as in the manuscript text (two occasions).

- Figure 7:
  Symbol explanations were added in the caption for the $\delta^{234}U$ plot to distinguish our new data from the measurements of Shakun et al., 2007.

- Figure 8:
  The description of the Younger Dryas (YD) was moved to the end of the first sentence in the caption to clarify the meaning of the grey area.

**Response to Reviewer #2**

1. General comment on the clarity of the text.

   The entire manuscript was revised by a native speaker to increase the linguistic quality of the text and the clarity of formulations.

2. General comment on the usage of the terms "DCF" and "reservoir age":

   We thank the reviewer for pointing out this inconsistency. We have decided to follow the advice of the reviewer to refrain from using the term "reservoir age" for quantifications and changed the text to refer to "DCF". An additional sentence to clarify the usage of the term DCF and clarify the distinction between the two terms was added in section 4.2.

Specific comments:

- Lines 59-60:
  We have rewritten this sentence to clarify the meaning and make it more comprehensible.

- Line 164:
  For the duplicates, all steps including subsampling, chemical processing and measurement were repeated at the respective depths in the stalagmite. This information was added to the text.

- Line 168:
  The word "absolute" was replaced by "U-Th modeled"

- Line 196:
  The word "at" was added before "~ 11 kyr BP"

- Line 198:
  The word "of" was added between "factor" and "2".

- Lines 200-207:
  The original section was substantially rephrased to concisely highlight the connections between growth rate and stalagmite morphology as described by previous studies (e.g. Dreybrodt and Kaufmann). Increased stalagmite diameter and growth rate are caused by increased temperature and drip rate. However, for a time with most likely higher drip rates due to increased precipitation a higher growth rate is observed but stalagmite diameter has decreased. This is in partial conflict with the studies mentioned above, which is what this paragraph now highlights.
  After careful consideration, we believe this paragraph provides an important contribution to future studies focussing on stalagmite formation models, and have therefore decided to keep this rephrased version of the paragraph in the discussion.
  We hope this finds the agreement of the reviewer.

- Line 286:
  The term "secular variability" was replaced here and later in the text to clarify the meaning.

- Line 298:
  We thank the reviewer for pointing this out. We have corrected the incorrect word "temperate" and replaced it by "more humid" to include the mentioned studies. We think it is crucial to keep these references in the discussion because they highlight the large discrepancy of the mechanism

connecting calcite dissolution systematics and soil infiltration for humid conditions as opposed to a more arid climate zone in this study. Especially the significance of soil-processes on the rock dissolution control and subsequent dead carbon incorporation is of great importance for our study. We hope this corrected sentence is found to be adequate for the manuscript.

**Additional Changes by the Authors**

- Line 6: Jens Fohlmeister's affiliation was updated to read "Potsdam Institute for Climate Impact Research, Potsdam, Germany"

All contributors of this study have stated their agreement with the aforementioned changes and additions.

We again would like to thank the reviewers for their input.

Kind regards,

Steffen Therre, on behalf of all authors